# Inferring causal connectivity from pairwise recordings and optogenetics

**Mikkel Elle Lepperød**[1,2]*, **Tristan Stöber**[2,3,4], **Torkel Hafting**[1], **Marianne Fyhn**[2,5], **Konrad Paul Kording**[6]

**1** Institute of Basic Medical Sciences, University of Oslo, Oslo, Norway, **2** Simula Research Laboratory, Oslo, Norway, **3** Institute for Neural Computation, Faculty of Computer Science, Ruhr University Bochum, Bochum, Germany, **4** Epilepsy Center Frankfurt Rhine-Main, Department of Neurology, Goethe University, Frankfurt, Germany, **5** Department of Biosciences, University of Oslo, Oslo, Norway, **6** Department of Neuroscience, University of Pennsylvania, Pennsylvania, United States of America

* mikkel@simula.no

**Data Availability Statement:** All code used to produce data and results can be found at https://github.com/lepmik/causal-optoconnectics.

**Funding:** This research was partly funded by the National Institutes of Health (NIH) grant

## Abstract

To understand the neural mechanisms underlying brain function, neuroscientists aim to quantify causal interactions between neurons, for instance by perturbing the activity of neuron A and measuring the effect on neuron B. Recently, manipulating neuron activity using light-sensitive opsins, optogenetics, has increased the specificity of neural perturbation. However, using widefield optogenetic interventions, multiple neurons are usually perturbed, producing a confound—any of the stimulated neurons can have affected the postsynaptic neuron making it challenging to discern which neurons produced the causal effect. Here, we show how such confounds produce large biases in interpretations. We explain how confounding can be reduced by combining instrumental variables (IV) and difference in differences (DiD) techniques from econometrics. Combined, these methods can estimate (causal) effective connectivity by exploiting the weak, approximately random signal resulting from the interaction between stimulation and the absolute refractory period of the neuron. In simulated neural networks, we find that estimates using ideas from IV and DiD outperform naïve techniques suggesting that methods from causal inference can be useful to disentangle neural interactions in the brain.

## Author summary

Understanding the interactions between neurons is essential for grasping brain function. Neuroscientists often face a challenge when stimulating multiple neurons simultaneously: determining which neuron influenced another. The use of optogenetics, where neurons are controlled with light, has improved precision, but when several neurons are activated at once, it's difficult to discern the specific source of influence. In this study, we discuss the potential pitfalls of overlapping influences. We introduce techniques from econometrics, namely instrumental variables (IV) and difference in differences (DiD), as a method to better identify neuron-to-neuron interactions. Our tests on simulated neural networks indicate that these techniques can be more effective than some traditional methods. By

R01MH103910 to KK. In addition, the Research Council of Norway Grant 231248, 300504 to TH and MF. KK received salary from the University of Pennsylvania, MEL, MF, TH TS received salary from the University of Oslo, MEL, TH, MF received salary from Simula Research Laboratory. The funders had no role in study design, data collection and analysis, decision to publish, or preparation of the manuscript.

**Competing interests:** The authors have declared that no competing interests exist.

integrating approaches from neuroscience and econometrics, we aim to enhance methods for understanding neural connections in the brain.

## Introduction

A central goal of neuroscience is understanding the neural mechanisms or causal chains giving rise to brain activity underlying perception, cognition, and action. Although simultaneously occurring processes at many spatial and temporal scales contribute to brain activity, one of the defining features of neural networks is their connectivity. Connectivity directs neuronal communication. This communication is causal; connection strength controls the causal effect when one neuron signals to another neuron. The human brain contains about 86 billion neurons [1] with countless connections. A complex system of this scale is incredibly hard to understand because of the numerous ways the contributing elements may interact internally [2]. While observing correlations within such systems is relatively easy, transitioning from observed correlations to a causal or mechanistic understanding is hard. For example, suppose we observe a given correlation between neurons A and B. In that case, this may be because of a causal interaction between A and B, because of common input (confounder bias), or as a side effect of correcting for other effects (collider bias). After all, the same activities may emerge from different and distinct causal chains [3, 4].

In today's typical studies in systems neuroscience, we only record from a small subset of all the neurons [5]. The data obtained from such recordings, e.g., from electrophysiology or *in vivo* imaging, is observational, as opposed to interventional involving perturbations. From observational data we generally cannot know how much of the observed activity was caused by observed or unobserved confounding. Unobserved neural activity confounds estimates of causal interactions and makes it challenging to reveal underlying mechanisms [6]. Relationships estimated from observational data in the presence of confounders may contain significant errors and lead to incorrect conclusions [7].

Confounding is the big threat to causal validity [8] irrespective of using simple regressions or advanced functional connectivity techniques [9–12]. Inference techniques, such as maximum entropy-based inverse Ising inference [13], and generalized linear models (GLMs) [14] may give solutions to confounding from observed variables as they support "explaining away" background activity [9]. However, this is only a meaningful strategy if most neurons are included in the recordings. Even a fully observed system is only identifiable under certain assumptions about nonlinearity or noise [15, 16]. For example, systematic errors can occur due to confounding emergent network states [17]. Moreover, fully observable scenarios are rare in experimental settings, especially in the mammalian brain with its many millions of neurons. Brain data will, therefore, rarely satisfy the criteria necessary for causal identifiability. The general conclusion from causal inference is that naïve regressions in partially observed systems will not reveal causality. We aim to circumvent this by developing new methods to identify effective connectivity between arbitrary neuron pairs.

In this paper, we restrict our focus to causal effects related to measuring effective connectivity. This is not just the direct influence of one neuron on a downstream neuron. Instead, it is defined as the change in postsynaptic spike probability caused by an extra presynaptic spike. Therefore, effective connectivity includes direct effects, bi-synaptic and other effects from the network. During perturbations, this effect is directly measurable. On the other hand, estimating the direct causal effect will generally require some level of modeling. Estimating the direct causal effect from the effective connectivity is an interesting and, to our knowledge, unresolved

problem in computational neuroscience. However, in many cases, effective connectivity is meaningful, e.g., when reconstructing a circuit with equivalent information flow. After all, a neuron will exert its effect through many paths. This paper thus focuses on effective connectivity, which we term causal influence.

To estimate the causal influence of one neuron on another neuron, stimulating the presynaptic neuron is the gold standard. Indeed, a standard definition of causality is the effect of an intervention changing one variable in the system, independently of all the other variables [8]. If we can experimentally stimulate and fully control single neurons, the ability to estimate causal relationships by regression is possible. However, gold-standard perturbations are experimentally challenging (it is challenging to stimulate just one neuron and get complete control) and may yield low cell counts because it requires intra- or juxtacellular [18, 19], or two-photon stimulation [20–26]. Therefore, it would be highly desirable to obtain causality from widefield one-photon optogenetic [27, 28] stimulation combined with neural recordings of large neuron populations. We focus on widefield one-photon optogenetic stimulation and refer to this simply as optogenetic stimulation.

Interpreting results from optogenetic stimulation in terms of causal interactions is often difficult. In most experimental settings, optogenetic stimulation will affect multiple neurons simultaneously. Hence, the stimulus will produce a distributed pattern of activity which then percolates through the network of neurons. Postsynaptic activity induced by stimulation could thus, in principle, come from any of the stimulated neurons, introducing problematic confounding.

For inspiration for resolving the confounding problem induced by optogenetic stimulation, we look to other disciplines addressing the confounding problem or, more precisely, confounded regressors. The inference of causality from observational data has been addressed in the fields of statistics [8], machine learning [4], and econometrics [7]. These fields have extensively worked on methods to estimate causality in the face of unobserved confounding and may provide clues for resolving these challenges in neuroscience.

A commonly used approach towards causal inference in economics is instrumental variables (IV), which has a genuinely long and exciting intellectual history [29–31]. The IV can be considered to emulate the treatment assignment in a randomized controlled trial. It is a measurable variable that directly affects one of the variables of interest and only indirectly affects the rest.

Let us consider a common problem in neuroscience, the estimation of the influence of a neuron with spike activity $X$ onto another neuron with activity $Y$. We want to estimate the causal effect $\beta$ from one neuron onto another. One approach would be to estimate the conditional mean function $E[Y|X]$ with regression. For simplicity, we can assume the linear model $Y = \beta X + U$. Here $U$ represents other factors in the brain other than $X$ and $Y$, that are measured. Because of the many neurons and high degree of connectivity in the brain, the regressor $X$ is confounded; it can be correlated to $U$ through some unknown functions $p, f$ $(p(X) = f(U))$. This relationship implies that the regression estimate of $\beta$ will not estimate the magnitude of causation from presynaptic to postsynaptic, but its association. Regression can thus generally not provide causal insights.

IVs are a way of getting around this problem. Let us say we have a stimulation signal $Z$, which we will call an instrument and that this instrument affects $X$. However, the setting is generally not that no stimulation means $X = 0$ and stimulation $X = 1$. Instead, realistic stimulation protocols would rather increase the probability of spiking $E(X|Z = 1) > E(X|Z = 0)$. If stimulation only adds say 1/10th of the probability of a spike (e.g. $p = .3$ instead of $p = .2$) then our estimates of causal effect sizes would be off by a factor of 10. The intuition behind the IV approach is that it estimates how much the stimulation affects X and then only uses this

stimulation-related aspect of X to identify the influence on Y. So, if the stimuli are random in time and uncorrelated with $U$, we may use $Z$ as an IV and obtain the true causal effect using the so-called Wald estimator [32] given by

$$\hat{\beta}_{IV} = \frac{\mathrm{E}[Y|Z=1] - \mathrm{E}[Y|Z=0]}{\mathrm{E}[X|Z=1] - \mathrm{E}[X|Z=0]}. \tag{1}$$

This formula is standard for IV identification found in textbooks [7, 33]; see S1 Appendix for proof of identification, both parametrically and non-parametrically. In other words, IVs are a trick to use upstream randomness to identify a causal system.

For an instrument to be "good", it must be unaffected by other variables, which is the case for single-neuron stimulation. However, the stimulation cannot be used as an IV for multi-neuron stimulation because it will drive parts of the rest of the brain dynamics, it will affect $U$, violating the key assumption of IV. In the brain, almost everything is affected by the network state. However, certain variables can be more or less affected. For example, the overall activity of the network is due to slow and strongly non-random dynamics. In contrast, the temporal pattern of when a neuron is in a refractory state may be in good approximation random. For example, if neurons are spiking according to conditional Poisson distributions, their exact timing conditioned on the network state will be random. While refractoriness may not be perfectly random, the exact spike-times are notoriously difficult to predict [9], suggesting that refractoriness is quite random. The absolute refractory period may serve as an IV because a neuron is randomly unaffected by stimuli during refractoriness.

When dealing with temporal data, we can also use an additional method commonly used for causal inference: difference in differences (DiD). The idea is that we have two time-series that are identical (or more specifically, parallel) apart from a perturbation affecting only one of them. For example, we may have two neural trajectories, one with perturbation and one without. By subtracting out the unperturbed one we can get rid of biases introduced by the ongoing dynamics. DiD in this sense is a way of producing a synthetic estimation problem that is less affected by biases (or unaffected if its assumptions are given).

For our stimulation problem we can formalize the effect of treatment $X$ in a longitudinal study with two time points $Y^*$, $Y$. We can calculate the differences $(\mathrm{E}[Y|X=1] - \mathrm{E}[Y^*|X=1]) - (\mathrm{E}[Y|X=0] - \mathrm{E}[Y^*|X=0])$ where the two differences in parenthesis denote a treatment group $X = 1$ and a control group $X = 0$, respectively. For example, we may measure baseline firing rates $Y^*$ in the primary visual cortex during standardized visual stimulation in control and treatment groups of animals over several trials to estimate $\mathrm{E}[Y^*|X=0]$ and $\mathrm{E}[Y^*|X=1]$. Then, after giving the treatment animal group $X = 1$ a drug, you measure both groups to obtain $\mathrm{E}[Y|X=0]$ and $\mathrm{E}[Y|X=1]$. The estimates obtained from the control group correct for any changes over time due to other factors than the treatment itself, e.g. animals have different hormone levels during pre and post-treatment; see Fig 1g for an intuitive sketch. It is also possible to separate trials from a single individual into control and treatment blocks and adapt the DiD framework. When estimating connectivity, we can compare pre- and post-spikes in otherwise matched brain states using DiD to improve causality estimates.

Here we show that combining the IV and DiD techniques allows for estimating the effective connectivity between neuron pairs under optogenetic stimulation. We first show how conventional optogenetic stimulation of neurons introduces confounding. This confounding effect is then simulated in a simple network of three neurons using a Binomial GLM approach. With this simple model, we show that by using the refractory period as an IV, we can distinguish between connected and unconnected neuron pairs. Combining the IV estimate with DiD, we correct for biased neuronal activity due to state differences before and after stimuli. We

compare these estimates with a naïve, although widely used, cross-correlation histogram (CCH) method that fails to distinguish respective pairs. Furthermore, we infer effective connectivity in a simulated network of randomly connected excitatory and inhibitory neurons with distributed synaptic weights. With this data, we compute errors of the IV and DiD methods and show that they are robust to different simulated network states. Furthermore, we analyze false positive and false negative errors and goodness of fit on synaptic weights with pairwise assessments using ordinary least squares fitting (OLS) and our IV/DiD combination.

The observed differences between the IV/DiD and the CCH estimates underline the importance of considering potential confounding when estimating connections based on neural activity measurements.

## Results

### Confounding

When we simultaneously stimulate many neurons and observe the activity of a postsynaptic neuron after stimulation, it is hard to know which subset of the stimulated neurons caused the activity in the putative postsynaptic neuron—after all they are simultaneously stimulated. To test inference methods and build intuitions in such situations we design a simulation tool as there are no, to our knowledge, any clean dataset containing both recordings and ground truth available.

We employ a linear-nonlinear cascade model, similar to spike response models with soft threshold [14, 34–36] in the form of a binomial GLM given by

$$p_{i,t+1} = P(M_{i,t+1} = 1 | M_{i,t}, M_{j,t}, M_{i,t-1}, M_{j,t-1}, ..., M_{i,t-H}, M_{j,t-H})$$

$$p_{i,t+1} = \sigma \left( \sum_{\tau=t-H}^{t} \left( M_{i,\tau} r(\tau) + \sum_{j \in \mathcal{N}(i)} W_{ij} M_{j,\tau} c(\tau) \right) - b + U_i(t) \right) \quad (2)$$

$$M_{i,t+1} \sim Bernoulli(p_{i,t+1}).$$

These equations describe a probabilistic model for a higher-order discrete-time Markov process, where the state of node $i$ at time $t + 1$, denoted by $M_{i,t+1}$, depends on the previous states of the node itself and its neighbors $j \in \mathcal{N}(i)$ up to a history length $H$, with timesteps of 1ms. The probability $p_{i,t+1}$ is calculated using a logistic (sigmoid) function $\sigma$ and Bernoulli-distributed random variables. The function $r$ represents refractoriness which is set to a large negative value after a spike representing absolute refractoriness followed by an exponential term representing relative refractoriness. The constant $b$ reflects baseline activity, and the coupling function $c$ represents response dynamics modeled by an exponential term; see Binomial GLM simulation for further details. We thus have a simulation that we can use to test our methods; for parameters see Table 1.

To illustrate confounding effects, we first simulated a network comprised of three neurons ($i \in \{A, B, C\}$) shown in Fig 1a. The neurons have a baseline activity and are additionally driven by Poisson-distributed excitatory and inhibitory inputs of a timescale of 10ms. Together, this simulates slow network effects in the background noise $U$. The interactions between neurons create a simple system to explore causality: A and B receive stimulation, but only B drives C ($W_{CA} = 0$, $W_{CB} > 0$). We can then ask how to statistically resolve that only B drives C, a task called causal discovery, and how strongly B drives C, a task called causal inference [4, 7].

To estimate the actual causal influence of a stimulated neuron on postsynaptic neurons, we need to distinguish the influence of one stimulated neuron from the influence of other stimulated neurons. This procedure is challenging since the two neurons receive the same inputs.

**Table 1. Simulation parameters.**

| Name | Fig 1 | Fig 2 | Fig 3 | Fig 4 | Fig 5 | Fig 6 |
|---|---|---|---|---|---|---|
| $\gamma_S$ | 5.0 | 6.0 | 6 | [0, 8] | [0, 8] | 3.0 |
| $\lambda_S$ | 50.0 | 50.0 | 50.0 | 50.0 | 50.0 | 50.0 |
| $\lambda_S^{min}$ | 10.0 | 10.0 | 10.0 | 10.0 | 10.0 | 10.0 |
| $\lambda_S^{max}$ | 200.0 | 200.0 | 200.0 | 200.0 | 200.0 | 200.0 |
| $\tau_{ex}$ | 10.0 | 10.0 | - | 10.0 | 10.0 | 10.0 |
| $\gamma_{ex}$ | 2.0 | [0, 7] | - | 2.0 | 2.0 | 2.0 |
| $\lambda_{ex}$ | 100.0 | 100.0 | - | 100.0 | 100.0 | 100.0 |
| $\lambda_{ex}^{min}$ | 30.0 | 30.0 | - | 30.0 | 30.0 | 30.0 |
| $\lambda_{ex}^{max}$ | 400.0 | 400.0 | - | 400.0 | 400.0 | 400.0 |
| $\tau_{in}$ | 10.0 | 10.0 | - | 10.0 | 10.0 | 10.0 |
| $\gamma_{in}$ | -5.0 | [0,-7] | - | -5.0 | -5.0 | -5.0 |
| $\lambda_{in}$ | 100.0 | 100.0 | - | 100.0 | 100.0 | 100.0 |
| $\lambda_{in}^{min}$ | 30.0 | 30.0 | - | 30.0 | 30.0 | 30.0 |
| $\lambda_{in}^{max}$ | 400.0 | 400.0 | - | 400.0 | 400.0 | 400.0 |

One intuition is to use unsuccessful stimulations, i.e. instances where one neuron is not responding due to membrane potential fluctuations, maybe a first suggestion which is fine if neurons are independent outside stimulation periods. However, membrane potentials can be correlated on long timescales such as with brain oscillations [37] or up-down states [38]. During these long timescale states, the network state confounds the effect of stimulations. Moreover, a correlation-based measure fails to discover the correct network structure. The CCH and OLS estimates described below show the effect on the stimulated neurons (Fig 1c). Since the stimulation affects both A and B simultaneously, it induces a strong correlation between A and B. This further confounds the system by rendering the correlation between B and C nonzero. When comparing the peak cross-correlations with baseline as described in Cross correlation histogram, specifically Eq (18), $\hat{\beta}_{CCH}$ estimates the connectivity of A and C to be nonzero (Fig 1d). A naïve reading of this result may suggest causal influences of both A and B on C [6]. This raises the question of whether there is any way to avoid this problem.

## Instrumental variables

Central to our approach is the concept of counterfactuals or potential outcomes, which allows us to compare what actually happened with what could have happened under different circumstances. Put simply, this means imagining what would have occurred if a subject had received a different treatment or intervention than what they actually received. We use this approach to estimate the causal effect of a given treatment or intervention on a specific outcome. Using this idea the causal effect is the difference $Y_i|_{do(T=1)} - Y_i|_{do(T=0)}$ where $Y_i|_{do(T=1)}$, shortened to $Y_i(T = 1)$, denoting the value $Y$ of subject $i$ given that we force some treatment $T = 1$, denoted by the *do* operator. For example, the treatment $T$ can indicate that a patch electrode gives a pulse $i$, and $Y_i$ indicates the increase in membrane potential. However, the causal effect is typically estimated by the average treatment effect (ATE) defined by the expected difference $E[Y(T = 1) - Y(T = 0)]$ since we can not observe the same identical trial with and without a given treatment.

In what follows, $X$, $Y$, $Z$ are random variables with binary values, and indexing refers to a trial, where we assume all trials are independent. Further, $X(Z = 1)$ refers to the value $X$ when $Z = 1$. We use $Z = 1$ to denote if an upstream spike preceded stimulus, and thus $Z = 1$ indicates

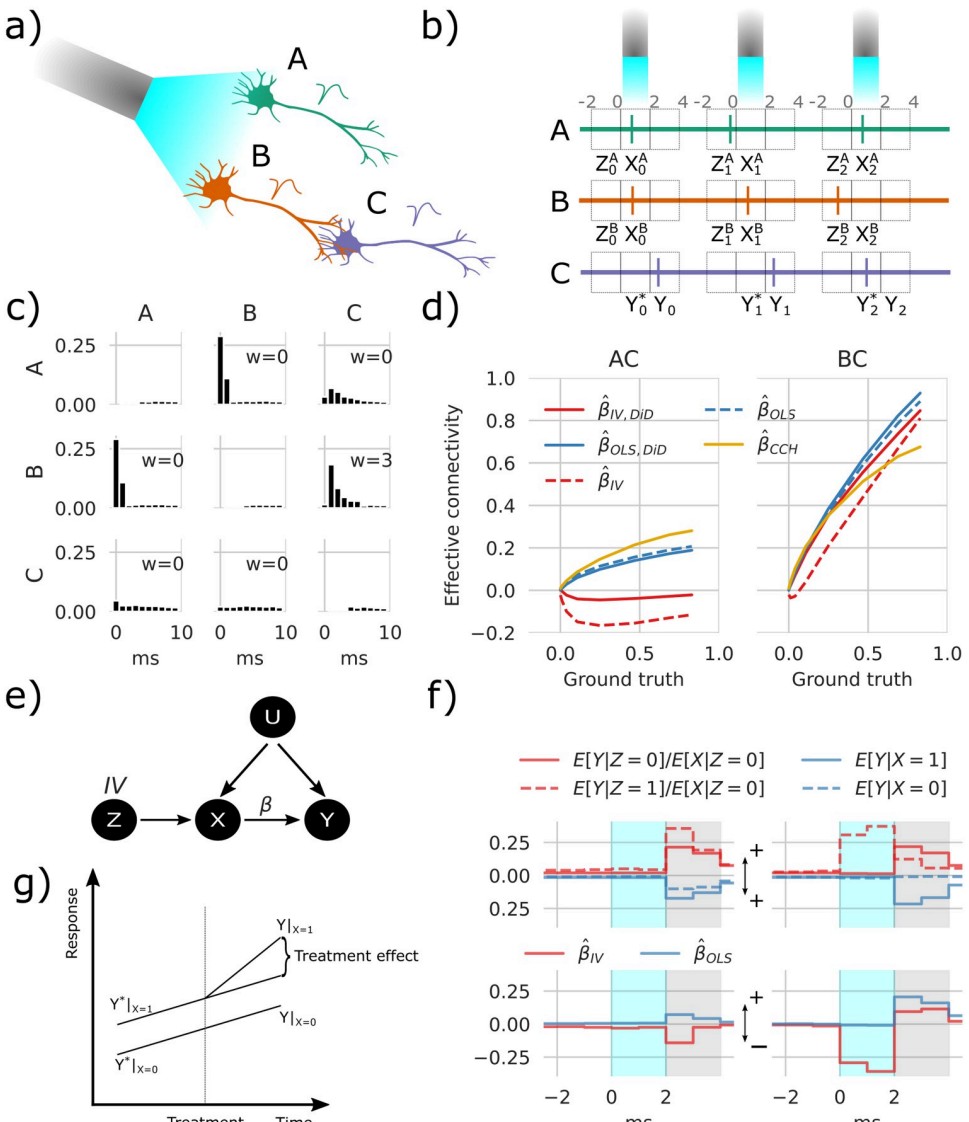

**Fig 1. Instrumental variables corrects spurious correlations induced by optogenetic stimulation. a)** Simple network containing three neurons shows stimulation configuration with blue laser light and the connections illustrated by axon touching dendrites. **b)** For each stimulation pulse, three spike detection windows are placed relative to stimulus onset time, pre, during, and post, which is stored as Z, X, and Y, respectively, where Z and X represent either A or B while Y represents C. **c)** Cross-correlation histogram (CCH) with the vertical axis showing the conditional expected value $E[M_i = 1|shift(M_j, \tau) = 1]$, $i, j \in \{A, B, C\}$, $\tau = 0, 1, \ldots, 9$ with shifts in timesteps $\tau$. **d)** Transmission probability estimates as a function of ground truth between AC, and BC. **e)** Directed acyclic graph (DAG) showing the relations between variables. The OLS method only uses $X$ to estimate $Y$, which fails because of the confounding factor $U$; the refractoriness denoted $Z$ affects $X$ without being affected by $U$ and can therefore be used as an IV. **f)** Conditional expectancies where the true weight = 7 rightmost values in d. The upper panel is mirrored around zero to aid visualization. The estimates in the lower panel represent differences between solid and dashed lines in the upper panel. The shaded gray area depicts the estimation window, and cyan shading indicates the stimulation window, which in this example also depicts DiD reference ($Y^*$). **g)** A sketch of the intuition behind DiD.

that $X = 0$ or, in other words, $X$ did comply with treatment. This is opposite to conventional notation, where $Z = 1 \rightarrow X = 1$, which might confuse readers with previous knowledge from causal inference. Here we want to obtain an estimate of $E[Y(X = 1) - Y(X = 0)]$ which we can write as $E[Y(X = 1)] - E[Y(X = 0)]$ when they are independent.

The IV technique has been used extensively in econometrics and can provide a provable causal estimate of $\beta$ given three main assumptions [33].

(i). Relevance: The instrument ($Z$) has a causal effect on the the regressor ($X$).

(ii). Exclusion Restriction: There must be no direct influence of the instrument ($Z$) on the outcome variable ($Y$); its effect must be fully mediated through the regressor ($X$).

(iii). Instrumental Independence: The instrument ($Z$) and the outcome variable ($Y$) must not share causes.

The validity of these assumptions is central when using the IV approach.

A final assumption **(iv)** is required to prove the causal validity of the estimator Eq (4), either a linear parametric assumption or a non-parametric assumption with effect homogeneity or monotonicity [33]; see S1 Appendix for proofs of identification following these assumptions.

## Refractoriness as an IV

One way to avoid confounding may be a timescale separation—using something that is random on a short timescale. Arguably, a neuron's refractoriness is such a variable. Suppose a neuron is in its refractory period (typically about 1–4ms, [39]). In that case, it will be less likely to fire, and during the absolute refractory period, no amount of stimulation could make it spike. Refractoriness gives us an interesting way of inferring causality by comparing stimulation of when a neuron can spike and when the neuron is muted. We would effectively not use the spikes but the missing ones for causal inference.

To think through confounding effects let us introduce some meaningful notation. Let $X, Y, Z \in [0, 1]^{N_S}$ denote random variables that indicate whether a neuron spiked relative to the stimulus during trials $i = 0, 1, 2, \ldots, N_S$. We evaluate pairwise relations: $Z$ and $X$ reference the spiking of one specific stimulated neuron at two time points, and $Y$ reference one specific downstream neuron. In other words, $X$ and $Z$ are extracted from one row ($M_i$) and $Y$ from another ($M_j$) given in Eq (2) using windows relative to stimulus onset. The window sizes were 2 ms with stimulus relative time given by -2,0,2 for $Z$, $X$, $Y$ respectively as shown in Fig 1b. As further depicted in Fig 1b, $Z_0 = 0$ indicates that the putative upstream neuron did not fire within a set time window before the first optogenetic stimulus, where subscripts indicate trial numbers. Furthermore, $X_0 = 1$ and $Y_0 = 1$ indicate the upstream and downstream neurons fired in this first trial. We seek to estimate whether a response $Y_0 = 1$ was caused by $X_0 = 1$. The putative upstream neuron is refractory if $Z_0 = 1$; in our simulations, the neuron fired 1–2ms before stimulus onset. With this notation we can now meaningfully address the problem.

We can be statistically precise about the causal reasoning. Consider the directed acyclic graph (DAG) [8] representing variables and their relations (Fig 1e). Here, $X$ represents a putative upstream neuron e.g., A or B, and $Y$ represents the putative downstream neuron e.g. C. The set $U$ contains unobserved network activity and the stimulus. During IV estimation, we use refractoriness which is (at least in first-order approximation) assumed to be random and influences stimuli response to infer the causal interaction between neurons. This approach is known to satisfy Pearl's criteria for causal inference [40], and IVs can reveal real causality.

In the following, we aim to estimate the effective connectivity defined as the difference in conditional probability $\beta_{ij} \coloneqq P(M_i = 1|do(M_j = 1)) - P(M_i = 1|do(M_j = 0))$. This means that the causal effect is defined as the difference in the post-synaptic neurons firing probability between a case where we set the pre-synaptic activity to one and the case where we set it to zero. Let us consider the case where there is no confounding, the neurons are conditionally independent

and we can simplify:

$$\beta_{ij} = P(M_i = 1 | M_j = 1) - P(M_i = 1 | M_j = 0)$$

Let us go through a concrete example. Let there be a network of two neurons $i$ and $j$. Let us assume $i$ is not refractory ($r = 0$) and a constant coupling $c = 1$. With only one spike from neuron $j$ after one time step and no noise ($U = 0$), we can derive $\beta_{ij} = \sigma(W_{ij} - b) - \sigma(-b)$. In this sense $\beta_{ij}$ is the causal effect induced by the underlying weight $W_{ij}$ and we can directly see that these two will be monotonously related. With refractoriness, however, we fit a linear model to represent the ground truth $\beta_{ij}$ and compute errors as the mean absolute error to estimate $E\left[|\hat{\beta} - \beta|\right]$; see Computing errors for further details. The case with refractoriness will be more complex, and sets up the problem we will be working on in the rest of this paper.

Let us start with the naïve estimate of $\beta$: the OLS estimate. For binary variables (allowing only $X = 0$ and $X = 1$), it can be written as the difference of conditional expectations [7]

$$\hat{\beta}_{OLS} = E[Y|X = 1] - E[Y|X = 0]. \qquad (3)$$

Note that here we implicitly condition on the stimulus due to stimuli onset selection when building the variables $X$, $Y$, $Z$. When $X$ is confounded (endogenous), i.e. $E[X|U] \neq 0$, the estimate $\hat{\beta}_{OLS}$ will be biased. What kind of biasing activities do we expect? Brains have slow dynamics, such as up-down states. Through those activities, pre- and post-synaptic neurons will be strongly correlated regardless of the actual synaptic connectivity. This consideration already presages the use of DiD below, which aims to minimize such biases. Importantly, OLS is an important baseline, used in many neuroscience settings.

The idea behind IV is to use a source of randomness that is local as a means of obtaining randomness. Here we want to use refractoriness $Z$ as an instrument. The IV-estimator for our binary case (see Wald estimator [32]) is given by

$$\hat{\beta}_{IV} = \frac{E[Y|Z = 0] - E[Y|Z = 1]}{E[X|Z = 0] - E[X|Z = 1]} \qquad (4)$$

Notice the flipped $Z$ compared to Eq (1) as it now represents refractoriness, not stimuli that further lead to $E[X|Z = 1] = 0$ by construction. If refractoriness would be entirely random we would eliminate bias and we may hope that, even for slight violations of this assumption, it will lower biases.

We can now investigate if the use of an IV (Eq (4)) gives a better estimate of connectivity strength compared to Eq (3) or simply analyzing the lagged correlations employing CCH calculated with Eq (18). Note that this CCH estimator can be considered an OLS estimate not conditioned on the stimulus. We use the IV estimator on the three-neuron system simulated with Eq (2) illustrated in Fig 1a. Interpreting the estimates of $\beta$, the IV converges to the ground truth for $\beta_{CB}$ but displays a slight negative bias in estimating $\beta_{CA}$ (Fig 1d red dashed lines). Contrary, the OLS and CCH methods both falsely conclude that $\beta_{CA} > 0$ (Fig 1d blue dashed lines represent Eq (3), and yellow lines Eq (18)). The technique based on refractoriness reduces bias but, in practice, does not abolish it.

When inspecting the stimulation response as shown in Fig 1f, we can see after splitting up Eq (4) (with $E[X|Z = 1] = 0$) and Eq (3) that the baseline values before stimulus-response are non-zero. The network dynamics in $U$ affect both neural refractoriness and downstream activities. To correct for these effects, we combine the approaches with a DiD correction [41]; see Difference in differences. We introduce superscript $*$ to indicate that the response window (Fig 1b) is shifted one window size backward in time. The OLS/DiD estimate is then given by

Eq (3) with $Y \to Y - Y^*$ i.e.

$$\hat{\beta}_{OLS,DiD} = E[Y|X = 1] - E[Y^*|X = 1] - E[Y|X = 0] + E[Y^*|X = 0] \tag{5}$$

and similarly, for the IV/DiD estimate given by

$$\hat{\beta}_{IV,DiD} = \frac{E[Y|Z = 1] - E[Y^*|Z = 1] - E[Y|Z = 0] + E[Y^*|Z = 0]}{E[X|Z = 1] - E[X^*|Z = 1] - E[X|Z = 0] + E[X^*|Z = 0]} \tag{6}$$

The DiD corrects for both the ongoing change and the network state and we should expect it to further lower bias relative to the IV. The IV/DiD method converges to the correct causal conclusions that $\beta_{CB} > 0$ and $\beta_{CA} \sim 0$ (Fig 1d solid red lines), with improved accuracy compared to the IV method. The IV/DiD method produces good estimates of the causal interactions between neurons for the simple three-neuron system.

## The strength of neural dynamics characterized by the condition number affects errors

Interacting neurons in a biological network exhibit inhibition and excitation, and their interplay produces nontrivial dynamics. When estimating the influences between neurons, we effectively solve an inverse problem. Such inverses are typically ill-posed problems, in the Hadamard sense [42] meaning a solution 1) does not exist, 2) is not unique, or 3) is unstable. Assuming the two first conditions to be untrue, i.e., a unique solution exists, the remaining lack of stability is referred to as an ill-conditioned problem. In statistics, this indicates whether small changes in (input) data have a comparatively small change in the (output) statistic and is meaningfully measured by its variance. When a statistical problem is ill-conditioned, the computed estimate may be far from the true estimate, regardless of our choice of estimator that minimizes the variance. This problem is typically due to multicollinearity. For example, in the linear model, $Y = \beta X + U$, multicollinearity is defined as near-linear dependence between column vectors in the design matrix $X$. In terms of neural measurements, $X$ would be a N-by-time matrix with columns representing population states at different timepoints, and multicollinearity would represent correlations between these population states. In the inverse problem $X'X\beta = X'Y$, with $'$ denoting transpose, the condition number $\kappa(X'X)$ is a measure of the existence of multicollinearity [43]. This condition number is given by the ratio of singular values of the Gram matrix $X'X$ by $\kappa(X'X) = \frac{\sigma_{max}}{\sigma_{min}}$. Large condition numbers imply that the inverse problem is ill-conditioned. Effectively, problems with large condition numbers, amplify a small amount of noise associated with small singular values, making them unstable. Multicollinearity may occur when a common, unobserved confounder drives neurons, as is very common in brains. Representing neurons as variables in the design matrix $M \in [0, 1]^{N \times T}$ (Eq (2)), the condition number was computed from the covariance of $M$ as the ratio of extreme singular values.

To assess how the proposed estimators performed under such circumstances, we simulated a fully connected random neural network of 200 neurons with an equal number of excitatory and inhibitory neurons (Fig 2a). Network dynamics were simulated with the Binomial GLM framework Eq (2). By stimulating five neurons, we examined the connection strength from these to the rest of the network and produced the results in Fig 2. The stimulus-trial onset times had a temporal Poisson distribution with a period 50 ms and were further clipped between 10–200 ms; see Binomial GLM simulation. The connectivity was given by a Gaussian distribution normalized by the square root of its size [44].

To compute errors we estimated a ground truth using a two-neuron network and computed the mean absolute error; see Computing errors. The error is thus interpretable as the expected

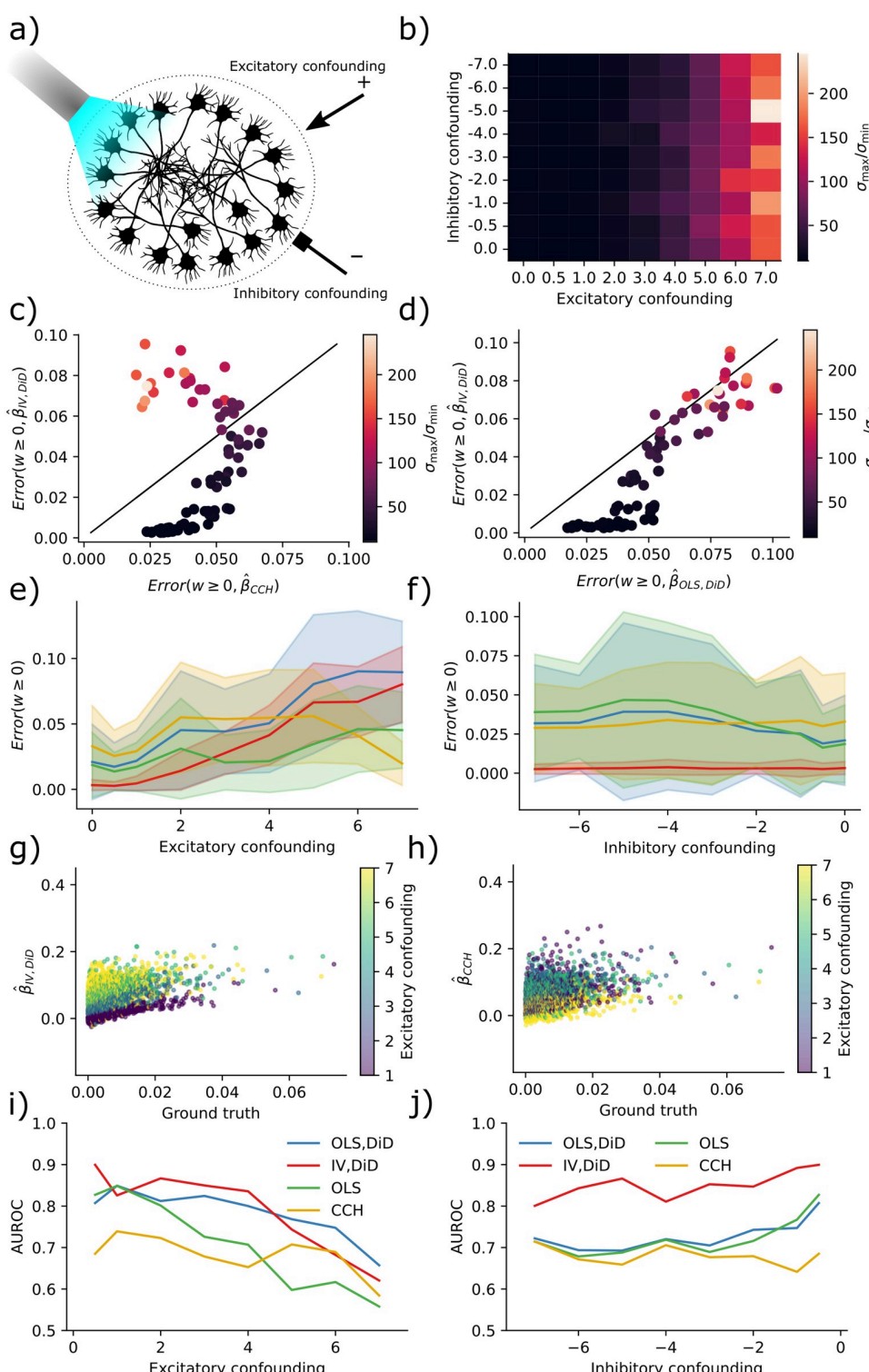

**Fig 2. IV is more robust to multicollinearity for moderate condition numbers. a)** Sketch of the neural network configuration. **b)** Condition number is more affected by excitatory drive than inhibitory drive. **c)** Error in IV/DiD and CCH for $w \geq 0$ and varying condition numbers as a function of drive strength. **d)** Error in IV/DiD and OLS/DiD for $w \geq 0$ and varying condition numbers as a function of drive strength. **e)** Errors as a function of the excitatory drive. Error bars represent one standard deviation of errors. **f)** Errors as a function of the inhibitory drive. Error bars represent one standard deviation of errors. **g)** Scatter plot of IV/DiD versus ground truth. **h)** Scatter plot of CCH

versus ground truth. **i)** AUROC as a function of the excitatory drive strength. **j)** AUROC as a function of the inhibitory drive strength.

absolute error performed by the estimator. Condition numbers were higher for larger drive strengths but less sensitive to inhibitory drive (Fig 2b). With higher condition numbers, errors increased in both the IV/DiD, and the CCH method for ground truth $\beta \geq 0$ where low condition number (dark points) means meaningful error for CCH but almost no error for IV (Fig 2c). The errors for IV/DiD were lower for moderate condition numbers and increased with increasing condition numbers (Fig 2d). The errors for IV/DiD were lower for moderate excitatory confounding (Fig 2e) but increased with confounding along with increased standard deviation measured across neurons illustrated with shading around the solid mean line. The errors for IV/DiD were constantly low with increasing inhibitory confounding (Fig 2f), with low standard deviations across neurons. Counter-intuitively, the error in CCH seems to get smaller with increased excitatory confounding, therefore we plotted the estimated connectivity against ground truth in Fig 2g and 2h. We observe that while the bias and variance increase for the IV/DiD estimator, for the CCH estimator the variance got smaller and the slope of increase also got lower, indicating that it is close to zero for all estimates, which would explain the artificial decrease in error.

To assert how the estimators performed as binary classifiers, classifying whether there is a connection or not, we computed the area under the receiver operating characteristic (AUROC) curve. This procedure measures the quality of classification at various levels of thresholds. It indicates how well the estimator will function in a classification task where AUROC = 1 indicates a perfect classifier, and AUROC = 0.5 indicates chance level. Classification accuracy was generally higher for IV/DiD compared to other estimators (Fig 2i) with excitatory confounding. Note also that the AUROC is consistently lower for CCH, which indicates that the low error seen for high confounding is indeed artificially induced. Classification accuracy was higher for IV/DiD compared to other estimators (Fig 2j) with inhibitory confounding. The network state can act as a confounder, and the condition number is a major way of indexing the confounding level.

## Stimulation, connection strength, and network size affects condition number and errors

To evaluate the IV/DiD method in a different setting, we simulated 480 networks with alternative stimulation strengths, number of neurons, and connectivity strength while keeping the excitatory and inhibitory drive constant. The connectivity was given by a Gaussian distribution normalized by the square root of its size [44] to keep the largest singular value constant to changes in network size.

By stimulating five neurons and examining the connection strength from these to the rest of the network, we produced the results in Fig 3. The condition numbers now correspond to network size, stimulus strength, and connection strength (Fig 3d). With higher condition numbers, the IV/DiD method outperformed the CCH method for ground truth $\beta \geq 0$ (Fig 3a). The IV/DiD method was more robust and was generally better than OLS/DiD (Fig 3b). The errors for IV/DiD were significantly lower than all other methods (Fig 3c). Condition numbers were higher for stronger stimulation and connection strengths and were more sensitive to these variables in larger networks (Fig 3d).

To see how the quality of the estimates varied with stimulus and connection strength, we plotted their relations with error bars showing a 95% confidence interval over the non-

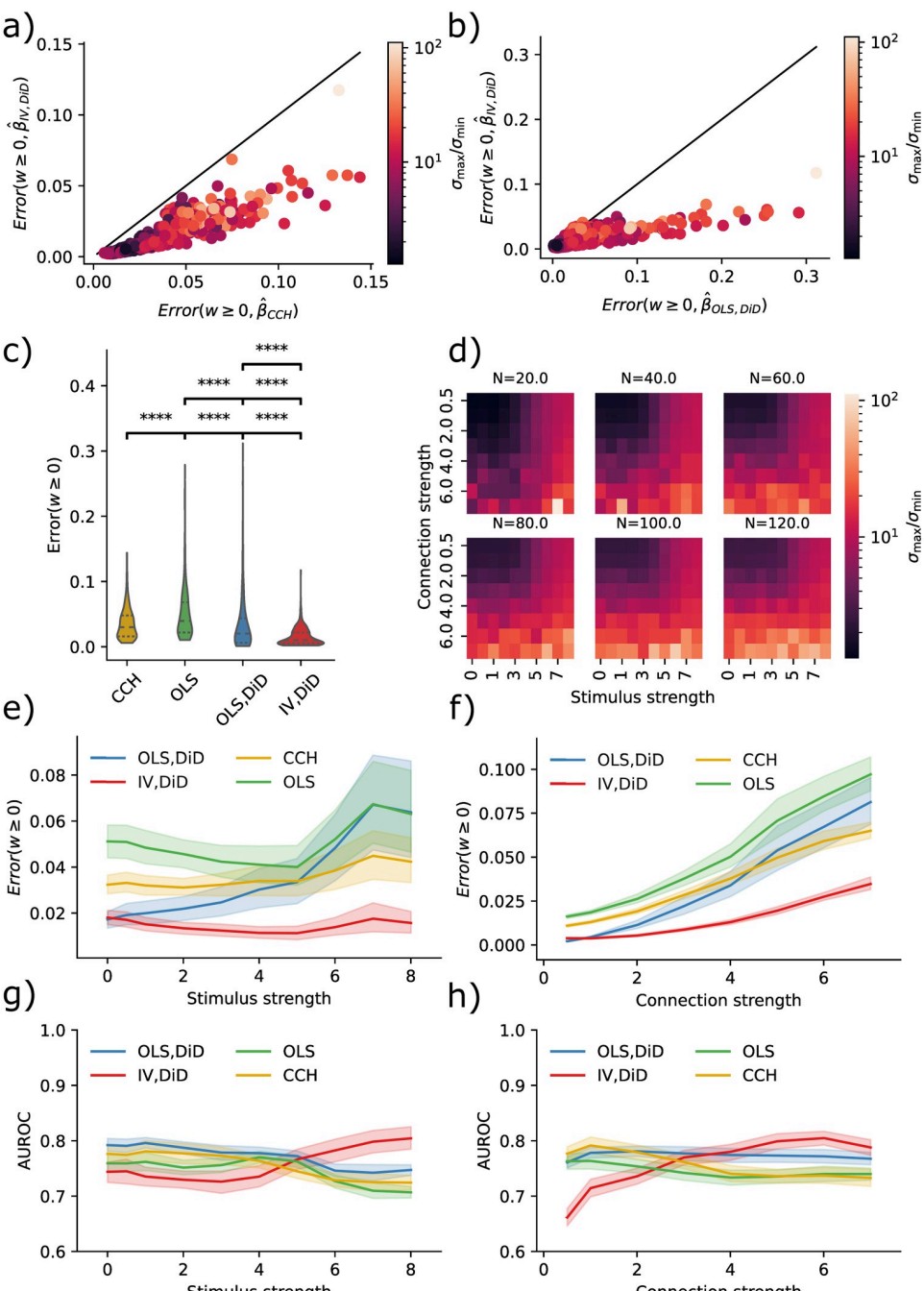

**Fig 3. Errors and corresponding condition numbers are affected by network size, stimulation and connection strengths. a)** Error in IV/DiD and CCH for $w \geq 0$ and varying condition numbers determined by stimulus strength, connection strength, and the number of neurons. **b)** Error in IV/DiD and OLS for $w \geq 0$ and varying condition numbers determined by stimulus strength, connection strength, and the number of neurons. **c)** Errors are significantly lower in IV/DiD than all other estimators; each line represents a comparison with the estimator corresponding to the left edge. **d)** Condition number as a function of stimulus strength, connection strength, and the number of neurons, color-coded according to the color bar given in (b). **e)** Errors from IV/DiD get lower with stronger stimuli than those in OLS, and CCH. **f)** Errors from IV/DiD increase with stronger connections, although less than OLS, OLS/DiD and CCH. **g)** AUROC from IV/DiD gets higher with stronger stimuli than those in OLS, and CCH. **h)** AUROC from IV/DiD gets higher with stronger connections, whereas those in OLS, and CCH varies less. Significance values are encoded as ns not significant, * $p < 0.05$, ** $p < 0.01$, **** $p < 0.0001$ computed with Mann Whitney U test.

visualized parameters, i.e. for stimulus strength error bars come from connection strength and network size, in contrast to Fig 2. The IV/DiD method yields lower error for higher stimulation strengths (Fig 3e). Intuitively, we need relatively strong stimuli for the downstream neuron to be affected by the instrument, this might explain the smaller errors with stronger stimuli seen in Fig 3e for IV/DiD. The IV/DiD method gave better classification results as measured with the AUROC, with stronger stimulus and in general for stronger connection strengths (Fig 3g and 3h).

## Connectivity affects statistical power, stimulus percentage affects error

Causal Inference techniques obtain better causal validity at the cost of statistical power. When the number of synapses per neuron increases, estimating the weights should become harder even if we could record from all neurons. To quantify this, we simulated ten networks with 200 neurons without additional confounding factors where we reduced connectivity or equivalently increased sparsity (0% meaning fully connected 90% meaning 10% connectivity). We found that the error decreased as a function of sparsity in IV/DiD, and OLS (Fig 4a). To further quantify robustness, we also simulated ten networks with 200 neurons, where we increased the percentage of stimulated neurons. The error increased with an increased percentage of stimuli (Fig 4b) as would be expected when considering that the condition number increases with increasing stimulus percentage.

By plotting the estimates as a function of the ground truth and sparsity, it is evident that the error is due to a more extensive spread which for the IV/DiD method decreases with sparsity (Fig 4c) while not so much for CCH (Fig 4e). When increasing the stimulus percentage, the spread increases for IV/DiD, and the center of the estimates at $\beta = 0$ is shifted upwards. However, the slope seemingly remains steady (Fig 4d). For the CCH method, the spread increased with increased stimulus percentage upon reducing again, the slope went towards zero, and the center at $\beta = 0$ is shifted upwards (Fig 4f).

To reach asymptotic convergence for IV/DiD, a large number of trials is needed ($\sim 10^6$) as seen in Fig 4g, for larger sparsity, more trials are needed, but with the result of a smaller error. When increasing the stimulus percentage, the convergence curve flattens out while being shifted towards larger errors (Fig 4h).

The IV/DiD method introduced here for the estimation of causal effects requires considerable amounts of data. The applicability of the proposed method may be limited to cases where weights are sufficiently sparse, and the stimulation effect is sparsely distributed among neurons with strong stimuli responses. Based on these findings, the right domain of application for the IV/DiD method would not be densely stimulated *in vivo* mammalian cortex but rather systems that are smaller, more sparse, or scenarios where only a small number of neurons are stimulated.

## Estimation under distributed optogenetic stimulation strength

Results presented in Fig 4b show that a uniform distribution of strong stimuli across the excitatory population is detrimental for the estimation of connectivity, although estimation is better with IV/DiD compared to OLS,(DiD), and CCH. However, a uniform distribution of stimuli strength does not reflect the distribution of light intensity obtained with optogenetic stimulation. Optogenetic stimulation is generally seen as a perturbation that, by and large, affects neurons in proximity ($< 1mm$) of the light source, decreasing the effect with distance. However, this needs to be more accurate in terms of conceptualizing the spatial effect of stimulation, as it depends on multiple factors. Light intensity and opsin density are essential as more light and ion channels will cause a more substantial effect on each cell. Moreover, the number of

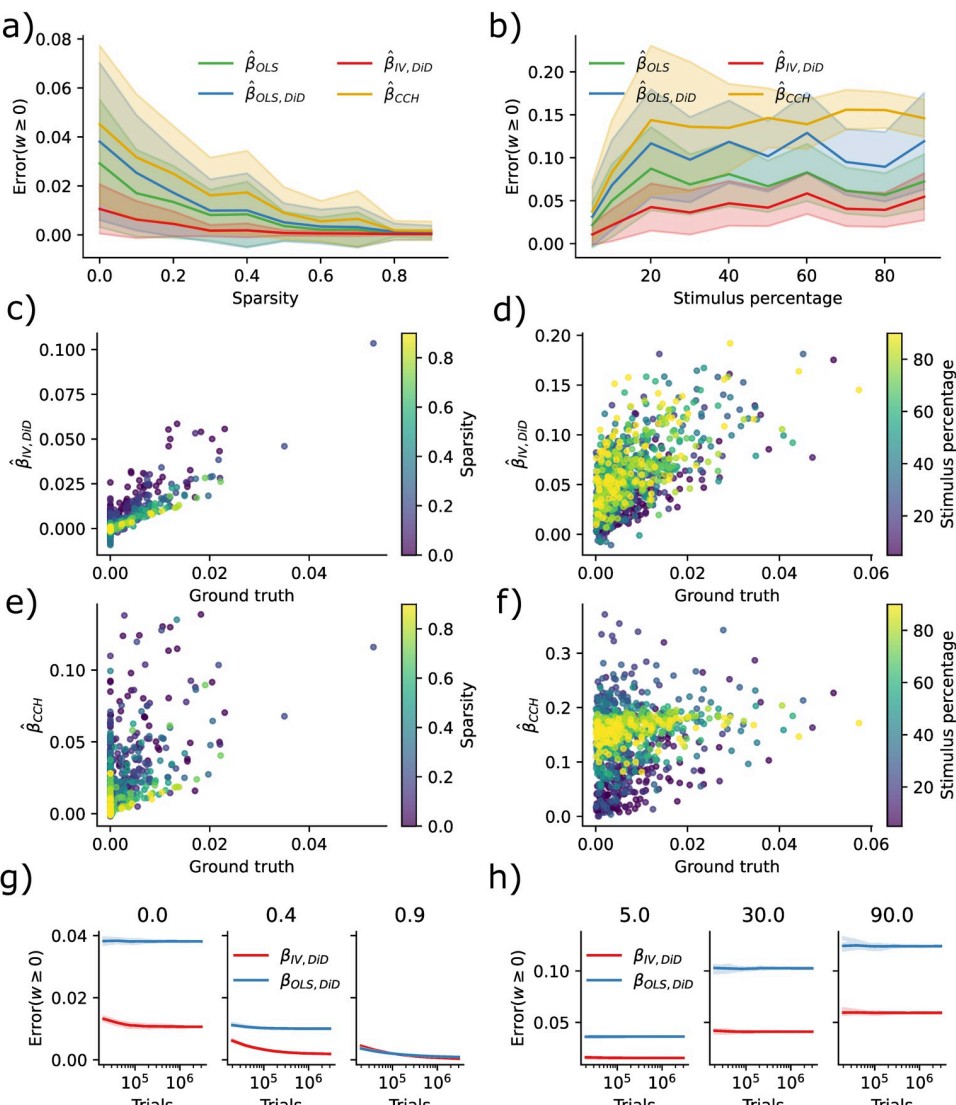

**Fig 4. Error diminishes with sparsity while is increased by increasing stimulus percentage. a)** Error as a function of sparsity. **b)** Error as a function of percentage of stimuluated neurons. **c)** IV/DiD estimates as a function of true weight and sparsity. **d)** IV/DiD estimates as a function of true weight and stimulus percentage. **e)** CCH estimates as a function of true weight and sparsity. **f)** CCH estimates as a function of true weight and stimulus percentage. **g)** Error as a function of the number of stimulus trials, sparsity is denoted in the panel titles. **h)** Error as a function of the number of stimulus trials, stimulus percentage is denoted in the panel titles.

potentially stimulated neurons is critical as more neurons will have a larger impact on the overall population activity. Finally, the physiological properties of the cells are important, as light may have a stronger effect on spiking activity when the cell's membrane potential is sufficiently close to the firing threshold. A relation between four parameters should give the induced effect of optogenetic stimulation as a function of distance: light intensity, the spatial distribution of neurons, distributions of membrane potential across neurons, and the distribution of induced photo-currents.

To estimate light intensity, we calculated the spatial extent of laser light delivered by fiber-optics under plausible experimental conditions according to [45]. To approximate an

optogenetic experiment we modeled the transmission of light through brain tissue with the Kubelka-Munk model for diffuse scattering in planar, homogeneous media [46] given by

$$T = \frac{1}{Sr + 1}. \tag{7}$$

Here $T$ denotes a transmission fraction, $S$ is the scattering coefficient for mice [45], and $r$ is the distance from a light source. Further, we combined diffusion with geometric loss assuming that absorption is negligible as in [45] and computed the intensity as presented in Fig 5a by

$$\frac{I(r)}{I(r = 0)} = \frac{\rho^2}{(Sr + 1)(r + \rho)^2} \tag{8}$$

where $r$ is the distance from the optical fiber and

$$\rho = \frac{d}{2} \sqrt{\left(\frac{n}{NA}\right)^2 - 1}. \tag{9}$$

Here $d$ is the diameter of the optical fiber, $NA$ is the numerical aperture of the optical fiber and $n$ is the refraction index for gray matter [46]; see numerical values for parameters in Table 2.

While the experiments in [45] uses blue light, their estimates assume no absorption making the equations somewhat general. This modeling of light intensity yields an approximately $1/r^2$ reduction with distance $r$ from the stimulation site (Fig 5a, cyan line). This is explained by the surface of a 3D shell growing with $4\pi r^2$ and photons will be roughly isotropic (but see [47]) beyond the scattering length as depicted in the inset of Fig 5a. The same number of photons must cross each sphere around the stimulation location unless absorbed or scattered inwards. As a result, the density of photons decreases with distance.

The number of illuminated neurons at a given distance will, however, increase with distance to the stimulation site, given that neurons are roughly uniformly distributed in brain tissue (Fig 5a, black line). It will increase by approximately $r^2$ with distance. This distance dependence derives from the same surface scaling for the 3D shell as the photon flow. Thus, the number of neurons that can be activated increases rapidly with distance.

To estimate the stimulation effect, the mechanism by which light affects spiking activity needs to be considered. The distribution of membrane potentials across neurons can largely characterize this. Surprisingly, this distribution has been observed to be symmetrically distributed and relatively flat [48–50]. The expected response from a pulse of light that induces a charge $Q$ should be proportional to the number of affected neurons whose membrane potential sits within a $Q/C$ range of the threshold ($C$ is the capacitance). Given that the distribution of membrane potentials is relatively flat (the density close to the threshold is generally within an order of magnitude of the density of its mode) suggests that the spiking response to a perturbation for any neuron is roughly proportional to the induced photo-current.

The peak amplitude of the photo-current relates approximately logarithmically to the light intensity. To estimate the peak amplitude photocurrent, we used the Hill equation fitted by parameters found in [51] given by

$$P = I_{max} \frac{I^n}{K^n + I^n} \tag{10}$$

Here, $I_{max} = 642 pA$ is the maximum current, $n = 0.76$ is the Hill coefficient, and $K = 0.84 mW/mm^2$ represents the half-maximal light sensitivity of the ChR2. We used the light intensity $I$ given by Eq (8) multiplied by an initial intensity of $10 mW/mm^2$.

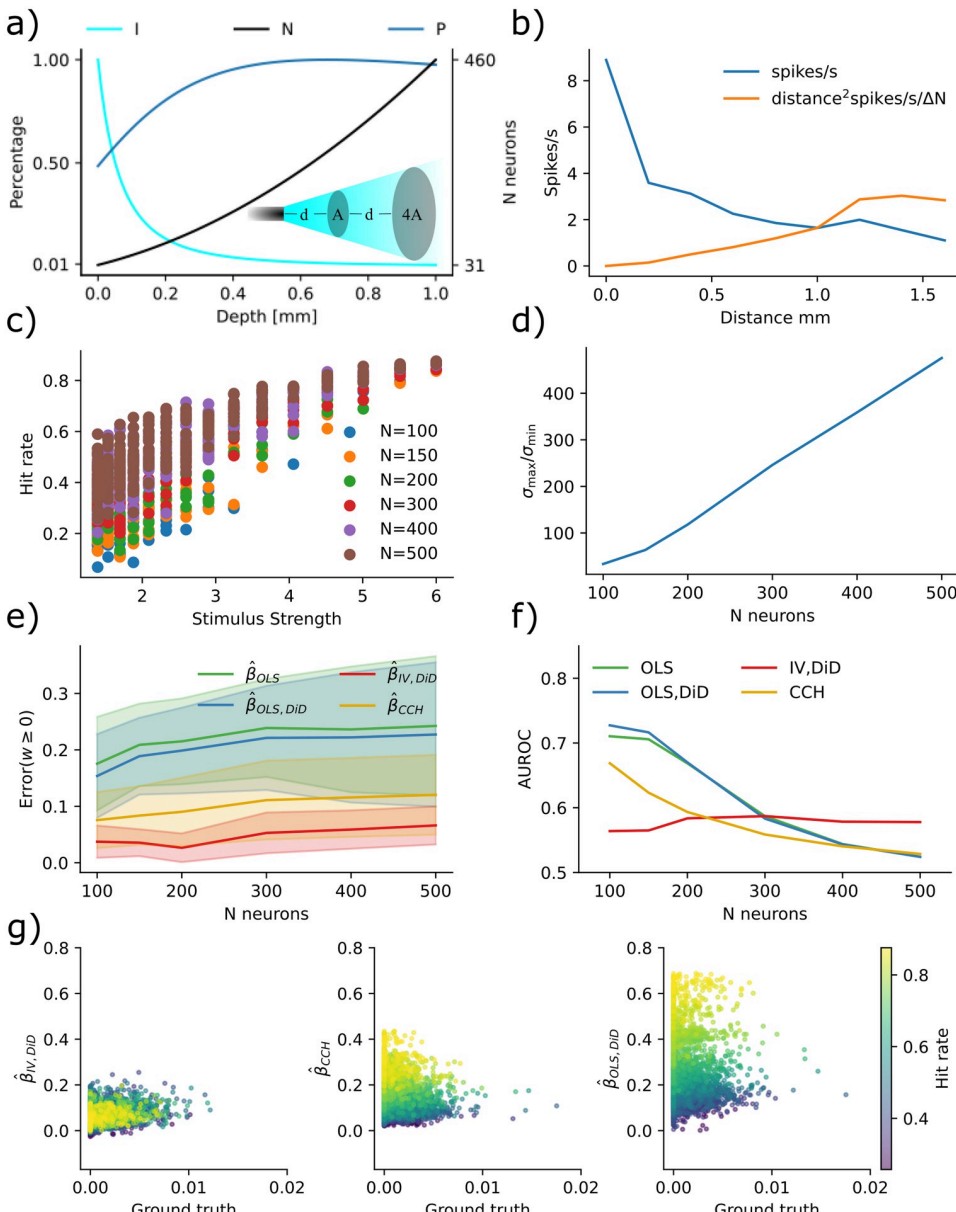

**Fig 5. Spatial extent of optogenetic stimulus. a)** Due to scattering and geometric loss the light intensity (I, cyan line) with an intensity of $10mW/mm^2$ exiting the optogenetic fiber follows approximately an inverse square law $r^{-2}$ where $r$ is the distance from the fiber. If neurons are uniformly distributed, the number of affected neurons in a spherical slice increases by $r^2$ (N, black line). The total photocurrent (P, blue line) summed over neurons in a spherical slice increased with distance due to the nonlinear relation between light intensity and photocurrent, depicted as a percentage of maximum. **b)** Experimental data for verification [52]. **c)** Hit rate increases relative to stimulus strength with network size. **d)** Condition numbers get severely large with increasing network size. **e)** Errors increase with network size. **f)** AUROC decrease with network size. **g)** Scatter of estimands to ground truth for N = 500.

Assuming that opsins are evenly distributed across neurons, the induced photo-current will not be proportional to light intensity—it will fall slower. Based on this, we calculate the overall stimulation effect as the product of the number of neurons in a spherical slice and the peak amplitude photo-current. This product increases with distance (up to the distance where absorption becomes important) (Fig 5a), blue line. In other words, there is more activation at

**Table 2. Stimulation parameters.**

| name | value | units |
|---|---|---|
| *density* | $\sim 1960$ | $Nmm^{-3}$ |
| *S* | 10.3 | $mm^{-1}$ |
| *NA* | 0.37 | |
| *r* | 0.1 | $\mu$m |
| *n* | 1.36 | |
| $n_{Hill}$ | 0.76 | |
| *K* | 0.84 | |
| *depth* | 0.7 | mm |

500um than at 100um—when neurons densely express opsins, optogenetics is not a local technique.

To verify this intuition with experimental data, we extracted data with WebPlotDigitizer (apps.automeris.io/) from [52] where an optic fiber was retracted while measuring average spiking activity (Fig 5b, blue line). By multiplying the distance squared with this activity, we get an experimentally valid estimate of the scaled population spiking activity (Fig 5b), orange line), which increases with distance. Thus, experimentally, optogenetic stimulation utilizing single photon activation does not produce a localized effect.

With this knowledge, we simulated more realistic distributions of stimuli strength to evaluate its effect on the proposed estimators. We simulated five networks of sizes ranging from 100–500 neurons with additional confounding factors and balanced excitation and inhibition. We gave each neuron a random spatial distance from the simulated optogenetic stimulus for stimulation. The stimulus intensity was then set according to Eq (10), and was constant throughout the trials. The trial onset had a temporal Poisson distribution with a period of 50 ms and was further clipped between 10–200 ms. To evaluate the stimulation effect, we measured the hit rate as the percentage of successful stimulation trials ($E[X]$). The hit rate was more prominent for larger network size and stimulus strength (Fig 5c). The effect of a larger hit rate with network size is likely due to more neurons being stimulated with high stimulus strength creating a cascade effect throughout the network. By increasing the maximum stimulus strength, we further observed a non-linear increase in hit rate across the population where at some point, almost all neurons had a hit rate at $\sim 90\%$. Condition numbers increased with larger network size (Fig 5d). Errors, as measured by the combined error across all hit rates, increased with larger network size (Fig 5e). Classifications were less accurate with larger network size (Fig 5f). Inspecting estimands as a function of ground truth, we observed that OLS/DiD, and CCH, to a large extent, predict the hit rate (Fig 5g).

## Inhibitory networks

In previous sections, we have only explored excitatory connections. However, the estimators could also be used with inhibitory connections. We simulated a network of 120 neurons with balanced excitation and inhibition while stimulating five inhibitory neurons to assess the effectiveness of inhibitory estimates. By examining the connectivity of these stimulated neurons, we generated the results presented in Fig 6. The IV/DiD and CCH are overall similar (Fig 6a) but are again outperformed by the OLS/DiD method (Fig 6b) for most variations of condition numbers. Comparing overall results, the OLS/DiD method was significantly better than the other method. Notably, contrary to estimating excitatory connections, stimuli benefited all estimators. Condition numbers were mainly affected by connection strength (Fig 6c). Errors

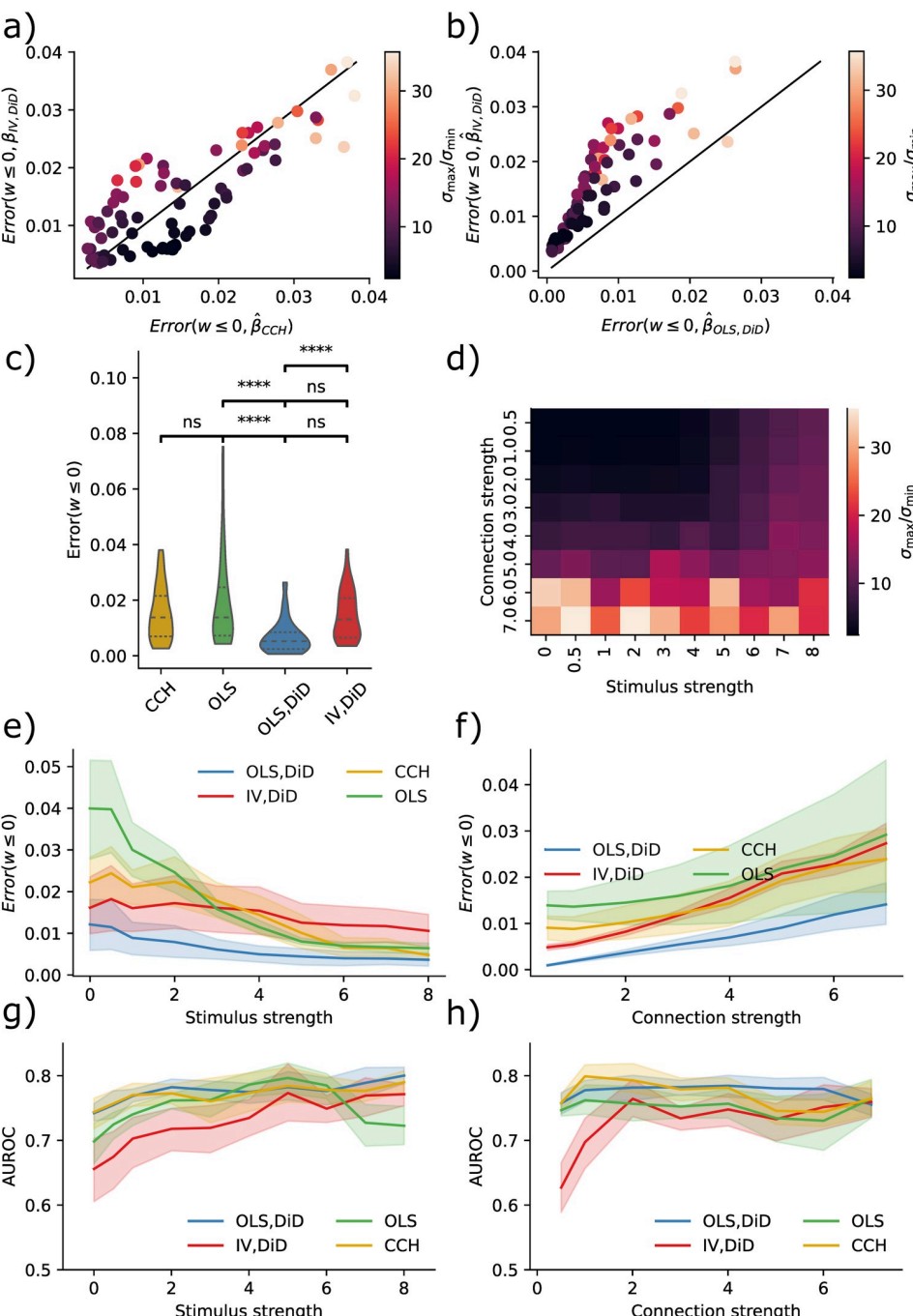

**Fig 6. IV is better than naive, but not OLS/DiD. a)** Error in IV/DiD and CCH for $w \leq 0$ and varying condition numbers as a function of stimulus strength, connection strength and number of neurons. **b)** Error in OLS/DiD and CCH for $w \leq 0$ and varying condition numbers as a function of stimulus strength, connection strength, and the number of neurons. **c)** Errors are significantly lower in OLS/DiD than all other estimators; each line represents a comparison with the estimator corresponding to the left edge. **d)** Condition number as a function of stimulus strength and connection strength, color-coded according to the color bar given in (a,b). **e)** Errors from OLS/DiD are generally lower with varying stimulation strength. **f)** Errors from OLS/DiD are generally lower with varying connection strength. **g)** AUROC from are comparable across methods with varying stimulation strength. **h)** AUROC from are comparable across methods with varying connection strength. Error bars represent 95% confidence across network size. Significance values are encoded as ns not significant, $^*$ $p < 0.05$, $^{**}$ $p < 0.01$, $^{****}$ $p < 0.0001$ computed with Mann Whitney U test.

were smaller for OLS/DiD for all connections, and stimulation strengths (Fig 6e and 6f). Despite giving poor effective connectivity estimates, classification was comparable across methods except for IV/DiD, which performed comparably worse (Fig 6g and 6h). Estimating the existence of inhibitory connections is relatively easy, but estimating their strengths appears harder.

## Discussion

Here we have asked if the refractory period of neurons can be used as an IV to reverse engineer the causal flow of activity in a network of simulated neurons. We have found that this approach performs considerably better than the naïve approaches. Moreover, we have found that neither a naïve linear regression model nor a naïve cross-correlation method produces reliable estimates of connectivity between neuron pairs. The IV approach improves the inference process by looking at the missing responses because refractoriness effectively allows better estimates of causal effects.

One might worry that if a downstream cell needs multiple simultaneous inputs to spike, and some of the stimulated cells converge onto the same downstream neuron, the estimated effective connectivity of upstream cell $X$ is biased outside of this stimulus regime. However, the stimuli are considered as part of the confounding error. It activates both the presynaptic neuron of interest and potentially many other presynaptic neurons. IV addresses this issue by removing the interaction between stimuli and postsynaptic neurons. Thus, the estimated effect of $X$ on $Y$ reflects the causal effect regardless of the stimuli. This is reflected in the error measure; zero error means that the causal effect is regardless of stimuli.

Moreover, every neuron is influenced by all of its presynaptic neurons. If the neuron is roughly linear then the causal effects $\beta_{ij}$ will approximate the weights $W_{ij}$. However, if neurons are very nonlinear then the real causal influences can not meaningfully be approximated by pairwise causal influences. As such, this approach, along with the bulk of the rest of the literature, is predicated on a largely untested assumption of neurons being approximately linear.

Currently, we have no ground-truth data set to test our technique and compare it with other approaches to actual neural activities. Ideally, we would have known causal effects from single-cell stimulation (e.g., from two-photon optogenetics) to establish causal effects. Such data should contain many randomly distributed, short, and intensive stimulation trials combined with traditional optogenetics, designed in a way where refractoriness matter. Since most optogenetic protocols use set stimulation frequency and light intensity, such a dataset, to the best of our knowledge, is currently not available and prevents us from testing how well our estimator would work on experimental data. Future experiments are needed to obtain reliable insights.

For the refractory period to be a good instrument, it must not be overly affected by the network activity. This criterion will be problematic in many cases. After all, network activity affects neuron activity and hence refractoriness. However, there are multiple scenarios where refractoriness will be a good instrument. For example, if we have balanced excitation and inhibition, we may expect largely independent refractory states of individual neurons. If a neuron biophysically implements something like conditional Poisson spiking, its refractory states will be random conditioned on the network state. Notably, the phase of a neuron may be far more random than the activity of the network as a whole.

The randomness of refractory times is the one factor that makes or breaks the IV approach we introduced here. Even if neurons' refractory states are strongly correlated during regular network operation, there may be ways of randomizing refractoriness. First, it would help to use a task and situation where neurons are as uncorrelated as possible. Second, we may use a

set of conditioning pulses of stimulation to increase the independence of refractory states. Giving one burst of stimulation strong enough to elicit multiple spikes from each neuron may effectively randomize their phases [53]. Third, we may utilize chemical, behavioral, or molecular perturbations to produce a good IV. For example, we can construct intracellular oscillators unaffected by neural activities or constructs that force a neuron into quiescence at random times. There has yet to be any effort to produce good IVs in neuroscience, so there may be many possibilities for improvements.

One popular way of estimating causal effects is fitting generalized linear models (GLMs) to simultaneously recorded neuron activities [14, 54]. GLMs are multiple nonlinear regressions and require multiple neurons to perform well. However, even if activity from all neurons were recorded, GLMs might fail to estimate causal connections [17]. However, complete recordings are not possible in the mammalian brain, especially not in primates, where recordings include only a minimal subset of the neurons involved in the actual computation. When using GLMs, one may accurately estimate latency distributions and sequences of spikes from individual neurons. These ideas should be merged with IV approaches. One of the strengths of the IV estimator presented here is that it only requires one pair to be recorded because we can utilize the randomness of the refractory periods along with random stimulations. Under those assumptions, the IV estimator can produce meaningful causal estimates.

The main problem with optogenetic stimulation, when used to infer connectivity, is its non-local property. This is due to the inverse relationship between changes in light intensity, and the affected number of neurons combined with a logarithmic relation between light intensity and photocurrent [51]. In addition, the distribution of membrane potentials across neurons is relatively flat [48–50], making neurons highly sensitive to perturbations. One could imagine situations where optogenetic activation was more local. If, for example, the membrane potential distributions were skewed with the mode far from the threshold, a powerful stimulus would be required for a neuron to elicit spikes. There could also be other ways of making optogenetic stimulation more local. For example, if one engineered opsins or brain tissue that are more light absorbent (e.g. by ubiquitously producing melanin), one could stimulate more locally. Having melanin under a ubiquitously expressed promotor in the brain would dramatically make optogenetics more local and could probably be a target for the construction of transgenic animals. How to engineer more localized stimulation is a significant problem when causally interrogating a system.

Very weak laser pulses in noisy networks might mainly elicit spikes in very few close-by neurons in each trial [55]. However, the stimulus will still affect the membrane potential of many neurons further away, some of which will spike. Therefore, more than weak stimulation is needed to remove the principal problem of correlation-based techniques. After all, the network still acts as a confounder, and, if anything, the weak stimulation will reduce the statistical power of the approach. Lowering stimulation amplitudes is not a way of obtaining meaningful causal estimates.

We acknowledge the potential advantages of two-photon optogenetic stimulation of arbitrary subgroups of cells *in vivo*; e.g. [56]. This technique can in some cases give more favorable and realistic conditions compared to blanket 1-photon optogenetic stimulation and holds the potential for application to a vast number of presynaptic and postsynaptic neurons across various brain regions [57]. Our intention is therefore not to undermine the utility of two-photon optogenetic stimulation. Rather, we aim to underscore some of its existing limitations, such as its inapplicability in freely moving animals or across different animal species for example due to the necessity for head fixation. Although limitations may be surmounted in the future, they currently present considerable challenges that constrain the applicability of this technology in

specific contexts. Consequently, we posit the necessity to persist in the exploration and enhancement of alternative approaches, such as 1-photon methods, which may be more apt for certain applications. This perspective underscores the necessity for a multifaceted approach to technological advancement in neuroscience, recognizing the strengths and weaknesses of each method to optimize their application in diverse research contexts.

There are many techniques for causal inference, most of which have yet to be applied to the field of neuroscience, and are based on approximating randomness in a seemingly regular world. In many cases, one could use regression discontinuity designs in a spiking system [58, 59]. Moreover, one could use a difference-in-difference approach [41]. Matching approaches [60, 61], can be used when comparing similar network states and their evolution over time. In general, neuroscience is in a quest for causal interpretations. Thus, we could benefit considerably by utilizing established techniques in the field of causal inference.

In the presence of ill-conditioning, the estimates of causal effects will be sensitive (unstable) to small changes in the underlying data. As outlined in [43], ill-conditioning can affect statistical analyses including the estimation of causal effects in three ways. First, measurement errors, such as a temporal shift in spike time estimate, e.g., low sampling frequency, inaccurate spike sorting, or general noisy measurement due to animal movement, can lead to significant estimation errors. Second, inference can give misleading results in ill-conditioning caused by bad design or sampling. There will always be natural variability in the observations, necessitating assessing ill-conditioning before any statistical analysis. Third, rounding errors can lead to small changes in data. Under ill-conditioning, this numerical problem is often not considered in neuroscience but will become evermore relevant when large-scale recordings require large-scale inferences. Causal inference approaches are not immune to the general problems coming from the ubiquitous ill-conditioning of neuroscience data.

The finding that inhibitory connections are very hard to estimate surprised us. After all, we had generally thought of excitation and inhibition as two sides of the same medal. However, we consistently find it hard to estimate inhibitory connection strengths. There may be some statistical reasons for this. An excitatory neuron can make another neuron fire at any point in time. An inhibitory neuron, on the other hand, can only prevent another neuron from spiking if it actually would be spiking. Neurons spend a vast amount of time not spiking. Similarly, neurons spend far more time being just below the firing threshold than being just above it. As such, the power to detect inhibition may be much higher than the power to detect excitation. Incidentally, this may also relate to the finding that there are far fewer inhibitory neurons in the brain than excitatory neurons: it is simply easier to learn how to excite than how to inhibit because there is more data.

Notably, practical short-term use of the techniques proposed exists. It is possible to stimulate just a local cluster of neurons optogenetically, say 10. This setup puts us into the situation of Fig 4b, where errors can be quite low, at least in the limit of relatively long recordings. As such, applying the methods introduced here to such experimental data, ideally with measured ground truth connectivity, is a promising and realistic next step.

## Methods

### Difference in differences

When dealing with temporal data we can use an additional method commonly used for causal inference, namely DiD. The effect of treatment in a longitudinal study with two time points $Y^*$, $Y$ is given by the average treatment effect of the treated (ATT) $E[Y(1) - Y(0)|X = 1]$. The ATT can be estimated by taking the differences $(E[Y|X = 1] - E[Y^*|X = 1]) - (E[Y|X = 0] - E[Y^*|X = 0])$ where the two differences in parenthesis denote a treatment group and a control

group respectively. The DiD estimate is valid given two assumptions: (i) consistency and (ii) parallel trends; see [7].

To sample the stimulus trials, let $t_i$ denote stimulus onset time for stimulus trials $i = 0, 1, 2, \ldots, N_S$, and spike response windows $[\tau_0^V, \tau_1^V]$ for variables $V \in \{X, Y, Z\}$ which are associated with spikes $t_k^V$ for upstream neurons $X$, $Z$ and downstream neurons $Y$. The variables are given by

$$V_i = \begin{cases} 1 & \tau_0^V \leq t_k^V - t_i \leq \tau_1^V, \ \forall t_k^V \\ 0 & \text{else.} \end{cases}$$

The spike windows used for DiD $[\tau_0^{V^*}, \tau_1^{V^*}]$ are given by $[\tau_0^V - \tau^V, \tau_1^V - \tau^V]$ for window size $\tau^V = \tau_1^V - \tau_0^V$ where $\tau_0^V < \tau_1^V$.

## Binomial GLM simulation

To model neuronal populations, we use a GLM framework similar to [35, 36]. This simulator and others alike are now implemented in Spikeometric spikeometric.readthedocs.io However, instead of using the Poisson GLM, we employed the binomial GLM, and instead of fitting the filters to neural data, we hand-tuned them. As discussed in the blogpost [62] the Poisson GLM comes with an exponential (unbounded) inverse link function which can produce nonphysical high spike rates. The binomial GLM comes with a (bounded) logistic inverse (sigmoid) link function which is ideal for spike simulations given as

$$\sigma(x) = \frac{\exp(x)}{\exp(x) + 1} \tag{11}$$

The generative model used throughout this paper is given in Eq (2).

Here the $N \times time$ matrix $M \in \{0, 1\}^{N \times T}$ indicated spikes at time $t_\Delta = \Delta tt$ where time bins were of size $\Delta t = 1ms$. Further, $H = 10$ denoted spike-history as the number of time steps backward. The bias $b = 5$ was hand-tuned such that average spike rates were $\sim 10Hz$.

The spike-indicator matrix $M$ was given by the sampled number of successes from the Bernoulli distribution with probability $p$.

The coupling matrix $W \in \mathrm{R}^{N \times N}$ contained cross-coupling weights. The cross-coupling filter was given by

$$c(t) = \begin{cases} \exp(-\alpha t) & \text{for } 1 \leq t \leq 5 \\ 0 & \text{for } 6 \leq t \leq H, \end{cases} \tag{12}$$

where $\alpha = 0.2$. The refractory filters representing the absolute and relative refractory period were given by

$$r(t) = \begin{cases} -100 & \text{for } t = 1, 2, 3 \text{ (absolute)} \\ -30 \exp\left(-\frac{1}{2}(t + 4)\right) & \text{for } 4 \leq t \leq H \text{ (relative)} \end{cases} \tag{13}$$

A normal distribution gave the connection strength $W \sim \mathcal{N}(0, \sigma/\sqrt{N})$ with zero mean and standard deviation scaled by the square root of the number of neurons. This scaling had two purposes: to keep the spike variance constant through time [44] and to keep the largest singular value constant as a function of network size [63].

The noise term $U$ was employed to impose perturbations upon the system. Noise terms were stimuli and excitatory and inhibitory external confounding given as $U(t) = U_S(t) + U_{positive}(t) + U_{in}(t)$ with respective strengths given by $\gamma_S$, $\gamma_{positive}$, $\gamma_{in}$. The simulated

optogenetic stimulation was given by

$$U_S(t) = \begin{cases} \gamma_S \text{ for } t_S \leq t \leq t_S + 2 \\ 0 \text{ else } . \end{cases} \tag{14}$$

The excitatory external confounding was given by

$$U_{ex}(t) = \begin{cases} \gamma_{ex} \text{ for } t_{ex} \leq t \leq t_{ex} + \tau_{ex} \\ 0 \text{ else} \end{cases} \tag{15}$$

and the inhibitory external confounding was given by

$$U_{in}(t) = \begin{cases} \gamma_{in} \text{ for } t_{in} \leq t \leq t_{in} + \tau_{in} \\ 0 \text{ else} \end{cases} \tag{16}$$

Stimulation onset times $t_S$, $t_{ex}$, $t_{in}$ were drawn from a clipped Poisson distribution with average period $\lambda_S$, $\lambda_{ex}$, $\lambda_{in}$ ms maximum period $\lambda_S^{max}$, $\lambda_{ex}^{max}$, $\lambda_{in}^{max}$ ms and minimum period $\lambda_S^{min}$, $\lambda_{ex}^{min}$, $\lambda_{in}^{min}$ ms.

The parameters in Table 1 were used to produce the resulting figures.

## Inhibitory and excitatory neurons

To follow Dale's law we split networks into equal amounts of excitatory and inhibitory neurons with the following trick. First, we drew random weights $W \in \mathrm{R}^{N/2 \times N/2}$ with zero mean and normalized by the square root of half of the number of neurons. Then we combined four variants of the weight matrix in the following way

$$W_{i,j} = \begin{cases} W_{i,j} \text{ if } W_{i,j} > 0 \\ \text{ else } 0 \end{cases}$$

$$W_{i,2j} = \begin{cases} W_{i,j} \text{ if } W_{i,j} > 0 \\ \text{ else } 0 \end{cases}$$

$$W_{2i,j} = \begin{cases} W_{i,j} \text{ if } W_{i,j} < 0 \\ \text{ else } 0 \end{cases}$$

$$W_{2i,2j} = \begin{cases} W_{i,j} \text{ if } W_{i,j} < 0 \\ \text{ else } 0 \end{cases}$$

The reasoning behind this procedure was to ensure we had a maximum singular value of one in the connectivity matrix which gives stable network dynamics.

## Distributed optogenetic perturbation intensity (Fig 5)

To estimate the distribution of light intensity on stimulated neurons, we distributed neurons uniformly in 15 spherical slices in the range $[0, 1mm]$, which had a radius given by the cone-shaped light; see Fig 1a inset.

Since the model neurons do not have membrane potentials, we scaled the maximum value of input current relative to the hit rate close to 100%.

## Cross correlation histogram

The statistical tests giving the probabilities $p_{diff}$ and $p_{fast}$ were done according to [55, 64]. We employed two tests to test whether the cross-correlation histogram (CCH) peak was significant. By using the Poisson distribution with a continuity correction [64] given by Eq (17), we calculated $p_{diff}$ by comparing the peak in positive time lag with the maximum peak in a negative time lag, called $p_{causal}$ in [55]. The probability $p_{fast}$ represents the difference between CCH and its convolution with a hollow Gaussian kernel [64]. These two measures of significance were required to be $< 0.01$ and given by

$$p(N|\lambda(m)) = 1 - \sum_{k=0}^{N-1} \frac{e^{-\lambda(m)}\lambda(m)^k}{k!} - \frac{e^{-\lambda(m)}\lambda(m)^N}{2N!}. \tag{17}$$

Here $\lambda$ represents the counts at bin $m$, and $N$ is the number of bins considered. To naively estimate the effective connectivity between pairs, we used the "spike transmission probability" defined in [55] as

$$p_{trans} = \frac{1}{n} \sum_{m=3ms}^{6ms} CCH(m) - \lambda_{Gauss}(m), \tag{18}$$

$n$ is the number of spikes detected in the presynaptic neuron and $\lambda_{Gauss}(m)$ is the CCH count convolved with a hollow Gaussian kernel at bin $m$.

## Computing errors

We computed errors according to

$$\begin{aligned}
\epsilon_{ij} &= \hat{\beta}_{ij} - \beta_{ij} \\
\beta_{ij} &= 0.9477(\sigma(W_{ij} - b) - \sigma(-b)) \\
\sigma(x) &= \frac{1}{1 + \exp(-x)}
\end{aligned} \tag{19}$$

The reported error was then given by the mean absolute error over all estimated weights

$$\text{Error} = \frac{1}{N} \sum |\epsilon| \tag{20}$$

We further separated error measures into excitatory and inhibitory connections denoted as $Error(w \geq 0)$ and $Error(w \leq 0)$, respectively. To estimate the ground truth $\beta$ we ran two blocks of simulations where two connected neurons; one with refractoriness given by Eq (13) and one with $r = 0$. In each block, we simulated $10^6$ time steps. With this setup we could verify that with $r = 0$ $\beta_{OLS}$ gave the correct estimate of $\beta = \sigma(W - b) - \sigma(-b)$. This gave us the confidence that we could use the OLS estimate with refractoriness to define the ground truth given in Eq (19). To do this we fitted a linear curve $y = \alpha(\sigma(W - b) - \sigma(-b))$ to the OLS estimate to obtain $\alpha = 0.9477$.

## Supporting information

**S1 Appendix.** The following proofs follow the book from Brady Neal [65]. Text A. Identification proof of ATE with linear parametric form of IV. Text B. Non-parametric identification of CACE with potential outcomes.
(PDF)

## Author Contributions

**Conceptualization:** Mikkel Elle Lepperød, Konrad Paul Kording.

**Data curation:** Mikkel Elle Lepperød, Tristan Stöber, Konrad Paul Kording.

**Formal analysis:** Mikkel Elle Lepperød, Tristan Stöber, Konrad Paul Kording.

**Funding acquisition:** Torkel Hafting, Marianne Fyhn, Konrad Paul Kording.

**Investigation:** Mikkel Elle Lepperød.

**Methodology:** Mikkel Elle Lepperød, Tristan Stöber, Konrad Paul Kording.

**Project administration:** Mikkel Elle Lepperød, Konrad Paul Kording.

**Resources:** Torkel Hafting, Marianne Fyhn.

**Software:** Mikkel Elle Lepperød, Tristan Stöber.

**Supervision:** Torkel Hafting, Marianne Fyhn, Konrad Paul Kording.

**Validation:** Mikkel Elle Lepperød, Tristan Stöber.

**Visualization:** Mikkel Elle Lepperød.

**Writing – original draft:** Mikkel Elle Lepperød, Konrad Paul Kording.

**Writing – review & editing:** Mikkel Elle Lepperød, Tristan Stöber, Torkel Hafting, Marianne Fyhn, Konrad Paul Kording.

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
