## [Decision Letter · Decision Letter 0]

27 Aug 2023

Dear Dr. Lepperød,

Thank you very much for submitting your manuscript "Inferring causal connectivity from pairwise recordings and optogenetics" for consideration at PLOS Computational Biology.

As with all papers reviewed by the journal, your manuscript was reviewed by members of the editorial board and by several independent reviewers. In light of the reviews (below this email), we would like to invite the resubmission of a significantly-revised version that takes into account the reviewers' comments.

We cannot make any decision about publication until we have seen the revised manuscript and your response to the reviewers' comments. Your revised manuscript is also likely to be sent to reviewers for further evaluation.

Sincerely,

Marcus Kaiser, Ph.D.

Academic Editor

PLOS Computational Biology

Lyle Graham

Section Editor

PLOS Computational Biology

Reviewer's Responses to Questions

**Comments to the Authors:**

Reviewer #1: The authors suggest the use of the idea of Instrumental Variables from economics to facilitate the inference of synaptic connectivity (effective connectivity?) from neural data where optogenetics can be employed as a perturbation. The analysis of the proposed method is done on simulated neural data showing the positive effect it has on the inference process.

I find the general idea of this paper quite interesting and important. Using perturbation (optogenetics or otherwise) as a tool for facilitating inference of connectivity is a very promising tool. Also, the concept of instrumental variables is a welcome novel addition to efforts for inferring connectivity. However, I find the results themselves weak and the exposition of the results quite unclear.

Regarding the results, my criticism are as follows

-the authors perform their analysis on simulated data. Although the authors do study the effect of various features of the simulated network (such as type external field), important features are unexplored. For example the network size is rather small (200 neurons: 100 excitatory, 100 inhibitory), the connectivity is stated to be excitatory. I would have excepted the simulations to be done with larger networks, and in particular with more realistic model neurons, e.g. Hodgkin-Huxely or Integrate-and-Fire. There many biologically realistic neural network models, e.g. of the visual cortex, and the authors come from one of the major centres for building such models. So I am a bit surprised that such analyses are not performed.

- The results are section 2.4 are based on stimulating 5 neurons. How does this change if the analysis is performed on a larger fraction of neurons being stimulated? Is it important to know a priori what the fraction of activated neurons are? if so, how can one have a good estimate of that in a real data?

- The authors do not explore their approach on any real dataset. It is true that for real datasets, the ground truth is not known. Still one can compare the results of different methods and see if they indeed do give different results, and how big the difference is.

Regarding the presentation, although I really liked the introduction, I found the presentation of the Results right after the Introduction without having the author know the main methodological approaches made the paper difficult to read. The Methods section at the end is also very cluttered and jumps between definitions, proofs and material (e.g. 4.3.2) that can also be described in the results section. Some of the material (e.g. proofs) can be moved to a supplemental information/apprendices. I think the paper needs a major rewrite even if not further analyses are described.

Reviewer #2: This study is interesting and creative, but unlike the authors I would not discount the usefulness of two-photon optogenetic stimulation of arbitrary subgroups of cells in vivo (Packer et al. 2015 https://www.nature.com/articles/nmeth.3217) to infer functional connectivity – it creates far more favourable and realistic conditions than blanket 1-photon optogenetic stimulation, and can already be applied to hundreds of presynaptic and thousands of postsynaptic neurons across brain areas (Fisek et al. 2023 https://www.nature.com/articles/s41586-023-06007-6) and more in the future. It would be useful to compare blanket 1-photon and patterned 2-photon stimulation when inferring functional connectivity.

My main concern, however, is that the simulated model system (eq. 2) used by the authors to apply their statistical methods to may not exhibit sufficiently realistic types of noise that are found in cortical networks in vivo, leading to an underestimation of confounds. London et al. (2010) showed that cortical networks are chaotic because small perturbations in the spiking history grow. This is because the probability of a presynaptic spike evoking an extra postsynaptic spike (which is on the order df a percent) multiplied by the fan-out of the presynaptic neuron (the number of its postsynaptic targets, which in the cortex is on the order of thousands) is much larger than one. Two spiking histories of the same network in response to the same stimulus will therefore be massively different if looked at in small time bins, which may interfere with the statistical analysis methods presented in this manuscript. Are the networks used in this manuscript sufficiently chaotic, and do perturbations in these models grow as fast (e.g. with a gain of 28, London et al. 2010) as they do in vivo? The authors should ensure and demonstrate that their model system generates sufficiently realistic irreproducibility of spiking histories of the network.

**Have the authors made all data and (if applicable) computational code underlying the findings in their manuscript fully available?**

Reviewer #1: Yes

Reviewer #2: Yes

PLOS authors have the option to publish the peer review history of their article (what does this mean?). If published, this will include your full peer review and any attached files.

Reviewer #1: No

Reviewer #2: No
---

## [Decision Letter · Decision Letter 1]

4 Oct 2023

Dear Dr. Lepperød,

We are pleased to inform you that your manuscript 'Inferring causal connectivity from pairwise recordings and optogenetics' has been provisionally accepted for publication in PLOS Computational Biology.

Best regards,

Marcus Kaiser, Ph.D.

Academic Editor

PLOS Computational Biology

Lyle Graham

Section Editor

PLOS Computational Biology

Reviewer's Responses to Questions

**Comments to the Authors:**

Reviewer #2: The rebuttals of the reviewers' comments by the authors are well argued, and the changes they made to their manuscript are good. Yet I'm torn because the changes are relatively minor and the authors did not include tests with larger networks as suggested by Reviewer 1, in which neurons have more realistic connectivity and synaptic weights. Future studies should involve random (Brunel) networks of integrate-and-fire neurons with realistic distributions of synaptic weights, as well as structured networks in which connectivity and connection strength represent similarity of sensory coding (see Cossell et al., 2015 for example).

**Have the authors made all data and (if applicable) computational code underlying the findings in their manuscript fully available?**

Reviewer #2: Yes

PLOS authors have the option to publish the peer review history of their article (what does this mean?). If published, this will include your full peer review and any attached files.

Reviewer #2: No

---

## [Editor Report · Acceptance letter]

30 Oct 2023

PCOMPBIOL-D-23-01040R1 

Inferring causal connectivity from pairwise recordings and optogenetics

Dear Dr Lepperød,

I am pleased to inform you that your manuscript has been formally accepted for publication in PLOS Computational Biology. Your manuscript is now with our production department and you will be notified of the publication date in due course.

With kind regards,

Zsofi Zombor
